# Farmers' Knowledge and Acceptance of Microalgae in Almería Greenhouse Horticulture

Ángela Ruiz-Nieto [1], Cintia Gómez-Serrano [1], Gabriel Acién [1] and Antonio J. Castro [2,*]

1  Department of Chemical Engineering, University of Almería, 04120 Almería, Spain
2  Andalusian Center for the Assessment and Monitoring of Global Change (CAESCG),
   Biology and Geology Department, University of Almería, 04120 Almería, Spain
*  Correspondence: acastro@ual.es

**Abstract:** Almería horticulture in SE Spain hosts the largest concentration of greenhouses in the world and faces important environmental sustainability challenges. Microalgae-derived applications are efficient nature-based solutions as they are used for wastewater regeneration or as biostimulants and biopesticides in agriculture. However, farmers' knowledge and acceptance of microalgae-derived applications remain unknown, which is a major barrier to its commercialization. This study explores current farmers' knowledge and acceptance of microalgae in Almería horticulture. Results revealed that there exists a significant lack of knowledge regarding the use of microalgae agricultural-based applications. Over sixty percent of farmers indicated that microalgae can have beneficial uses in agriculture, such as biostimulants or biofertilizers. However, although seventy percent of farmers expressed their willingness to use them, results also showed that only 32% of farmers using microalgae-derived applications have obtained satisfactory results. We call the urgent need for new communication strategies based on transdisciplinary approaches that increase farmers' knowledge around the multiple microalgae-derived products and applications in agriculture.

**Keywords:** microalgae; agriculture; biostimulant; biofertilizer; dryland; ecosystem services





## 1. Introduction

Globally, food consumption and trade are generating a high demand for agricultural systems and unprecedented pressure on natural resources, requiring a trade-off between food security and environmental impacts [1]. In Europe, the largest concentration of greenhouse horticulture is concentrated in the semi-arid coastal plain of the Almería province in southeastern Spain (Figure 1). Greenhouse production in Almería started after the 1960s and currently hosts the largest concentration of greenhouses in the world [1]. Since 1960, development strategies and a lack of land-use planning resulted in socioeconomic development in coastal areas that caused one of the most relevant land-use transformations in Europe [2]. The promotion of greenhouse horticulture has resulted in very important social and economic benefits for the province of Almería, besides having great negative impacts on native biodiversity and its natural resources posing new social challenges [3,4].

The annual economic contribution of Almería greenhouse horticulture is approximately EUR 1.8 billion and the related ancillary business sector generates another EUR 1.6 billion. In addition to the 15,000 family farmers engaged in productive activity, an additional 40,000 jobs are provided. This economic activity accounts for 40% of the province's GDP [5]. Currently, both small family farmers, as well as their cooperatives and related organizations and institutions will be in a position to carry out transformations in Almería to preserve the economic and social benefits that are positive, while gathering enough strength to face agroecological challenges [1,3,4].

Microalgae are produced in industrial facilities. When used for medium-value purposes, such as the production of agricultural products or animal feed, microalgae are

produced in open reactors such as raceways, more recently, thin-layer cascade photobioreactors were studied because of their higher areal productivity. One of the main advantages of producing photosynthetic organisms, including microalgae, is that they use sunlight as a source of energy and fix carbon dioxide to produce biomass and oxygen [6].

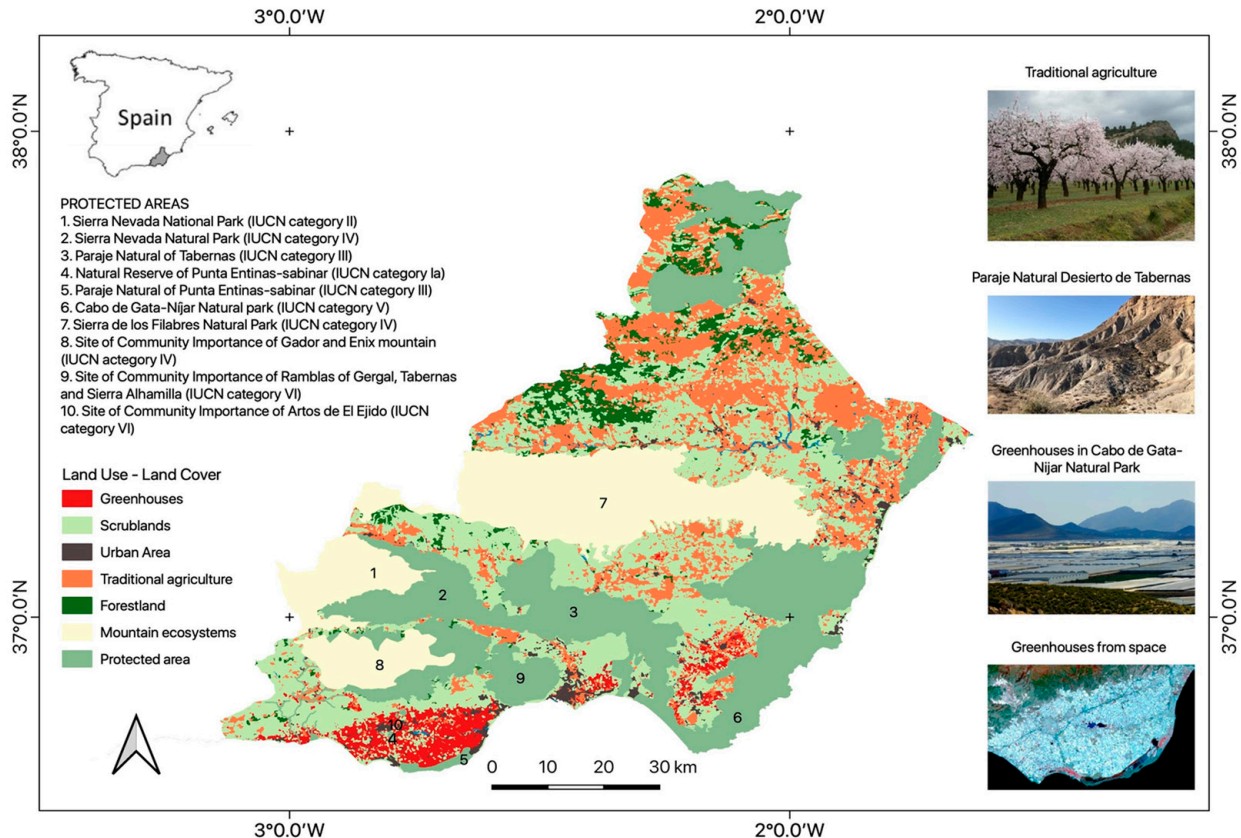

**Figure 1.** The geographic location of greenhouse horticulture in Almería, SE Spain.

Microalgae are becoming an important source of valuable products, especially for agriculture such as biostimulants, biopesticides or biofertilizers [6]. There is a growing interest in the application of natural products in agriculture that stimulate plant growth, increasing productivity without causing further environmental degradation (e.g., eutrophication, soil infertility and biodiversity loss) [7–9]. Biostimulants are defined as materials, other than fertilizers, that promote plant growth when applied in small amounts, increase mineral nutrient uptake and extend plant tolerance to abiotic stress, thus constituting an alternative to synthetic plant protection products [10,11]. The beneficial properties, from the point of view of agricultural applications of microalgae, result from their living conditions: permanent abiotic and biotic stress. This caused these organisms to develop mechanisms that protect them from drought, salinity, changes in light intensity, frost, and colonization by bacteria or fungi [11]. Commercially available microalgae-based products are already used for agriculture [12–14]. For the first time, microalgal extracts were used to stimulate not only plant growth but also plant mass, chlorophyll content, and carotenoid content and improve root development.

As mentioned above, the intensive horticulture model in Almería meets the growing demand for food from Europe with the consequent depletion of natural resources, but also the need to promote an agricultural model that helps mitigate the effects of climate change [14,15]. Therefore, it is a priority to develop commercial products from environmentally sustainable resources, such as microalgae-based products, which boost agricultural yields. Microalgae-based products are now extensively used to improve seed germination, plant growth, yield, flower set and fruit production, as well as post-harvest shelf

life [8,11,16]. Under the current climate change predictions in the Mediterranean, the Almería greenhouse horticulture is in urgent need of new innovative and sustainable ways that ensure the long-term sustainability of its agricultural production.

To push the use of this type of product among farmers it is necessary to know their current knowledge and acceptance on this type of solution. Using the case study of Almería greenhouse horticulture in SE Spain, this paper aims to assess farmers' knowledge and acceptance of the application of microalgae-derived products. To do so, using an online survey we first explored farmers' knowledge about microalgae's beneficial use and applications in greenhouse horticulture. Secondly, we specifically explored farmers' knowledge about the use of microalgae as biostimulants or biofertilizers, as well as the variety of crops in which they have obtained satisfactory results. Third, we explored farmers' acceptance of the diverse applications of microalgae in greenhouse horticulture. Finally, we explored and identified the main barriers farmers face in greenhouse horticulture as well as the ability of microalgae to mitigate them.

## 2. Materials and Methods

*Sampling Strategy and Survey Design*

We conducted a social sampling in the summer of 2020 with farmers of Almería's greenhouse horticulture (Figure 1). The sampling was conducted through an online survey (google forms) due to the COVID-19 pandemic. Farmers were randomly invited to respond to the survey through different social networks, including existing databases of the regional agencies that promote agri-food research initiatives such as Coexphal (Association of Fruit and Vegetable Producers' Organizations of Almería) and IFAPA (Institute for Agricultural and Fisheries Research and Training).

Once the invitation was accepted, farmers were first asked about their willingness to participate in the study and to respond to a survey focused on the environmental challenges of Almería's greenhouse horticulture. We informed them that all responses were anonymous, that the survey aimed just wanted to know their opinions, and that there were no right answers [17]. The survey included a total of 29 questions grouped in different sections to explore: (1) the farmer's knowledge regarding the concept of microalgae, their beneficial uses and applications; (2) the farmer's knowledge of the use of biostimulants derived from microalgae; (3) the main barriers that the use of microalgae can help to overcome in Almería greenhouse horticulture, and (4) the farmer's sociodemographic characteristics (see Appendix A). A panel illustrating different applications of microalgae applications in greenhouse horticulture was shown to help farmers to understand the goal of this study (see Appendix B). The original survey strategy was to carry out face-to-face surveys with farmers, either working in greenhouses or participating in agricultural cooperative initiatives; however, due to safety issues associated with the COVID-19 pandemic, surveys were conducted online.

First, a qualitative analysis was carried out with open questions on farmers' knowledge of the word microalgae (see Appendix A). To do so, we explored farmers' basic knowledge regarding the word microalgae. Second, a quantitative analysis was carried out to analyze farmers' knowledge of microalgae, their beneficial uses, species produced and microalgae-related applications in agriculture. Farmers' responses were grouped into major categories. Thus, microalgae's beneficial uses were grouped into six categories, including food for humans and animals, human health, environmental care, agriculture and biostimulants. Likewise, microalgae applications were grouped into four categories, including biostimulants, agriculture, environmental care and others.

We then used a Sankey diagram to unravel the relationship between farmers motivations to use microalgae in agriculture and potential microalgae applications. We used a panel illustrating specific applications of microalgae in agriculture based on scientific literature. Finally, to identify the potential barriers and opportunities associated with the use of microalgae, farmers were given a list of environmental barriers to indicate their

opinion on the importance of barriers, 0 being "It is not a barrier" and 10 meaning "It is an important barrier".

## 3. Results

A total of 62 farmers accepted the invitation and positively responded to the online survey. Of the 62 farmers, 39% and 31% were farmers located in the west and east of the Almería province, respectively, 23% belong to the municipality of Almería province, and the remaining 8% were from outside the province. Sixty-five percent of farmers belonged to a farming family, 40% owned agricultural land and 26% cultivated organic crops. Sixty-five percent were over 30 years old and 84% ran agricultural businesses with less than 10 workers (Table 1).

**Table 1.** Socio-demographic characteristics of farmers in Almería.

| Sociodemographic Variables | | |
|---|---|---|
| | Yes | No |
| Family of farmers | 65% | 34% |
| Land ownership | 40% | 60% |
| Organic farming | 26% | 74% |
| University studies | 50% | 50% |
| | Less than 30 | More than 30 |
| Age | 35% | 65% |
| | Less than 10 | More than 10 |
| Number of workers | 84% | 16% |
| | Male | Female |
| Gender | 50% | 50% |

### 3.1. Farmers' Knowledge Regarding Microalgae and Their Beneficial Use and Applications in Agriculture

Regarding farmers' knowledge of the word microalgae, we found a high diversity of responses. They were classified according to the following groups: marine plant (36%), sea (20%), food (14%), agriculture (7%), biostimulants (4%), environment (4%), examples of algae (4%), cosmetics and pharmacy (4%) and other answers that were not repeated (7%).

The analysis of the farmers' knowledge of microalgae's uses and applications indicated contrary findings. On one hand, we found that 66% of farmers identified microalgae as with capacity to provide many benefits (45% of farmers) or some benefits (21%) to agriculture. However, the results also indicated that 34% of farmers were not able to connect microalgae with any benefit or expressed their lack of knowledge about the benefits of microalgae (Figure 2A). Regarding the diversity of potential beneficial use that microalgae provide to agriculture, the results identified human health (27%), food for humans (24%) and environmental care (22%) (Figure 2B). Regarding the potential utility of microalgae in agriculture, 39% of farmers stated that microalgae have a great utility in agriculture, 31% stated some utility, and 30% of farmers either did not know any utility or indicated that microalgae have no utility in agriculture (Figure 2C). Finally, among the list of utilities identified by farmers, the majority identify agriculture (56%), biostimulants (28%), and environmental care (12%) (Figure 2D).

### 3.2. Farmers' Knowledge of Microalgae as Biostimulants and Biofertilizers in Agriculture

The question related to the use of microalgae as biostimulants or biofertilizers showed that 37% of the farmers did not know them, 31% were aware of them but have not used them, 27% were aware of and have used microalgae as both biostimulants or biofertilizers, and 5% of farmers have used microalgae only biofertilizers. No farmer had ever used microalgae only as biostimulants (Figure 3A). Those farmers who had successfully used microalgae as biostimulants or biofertilizers indicated they were more efficient in the production of tomatoes, pepper and cucumber (Figure 3B).

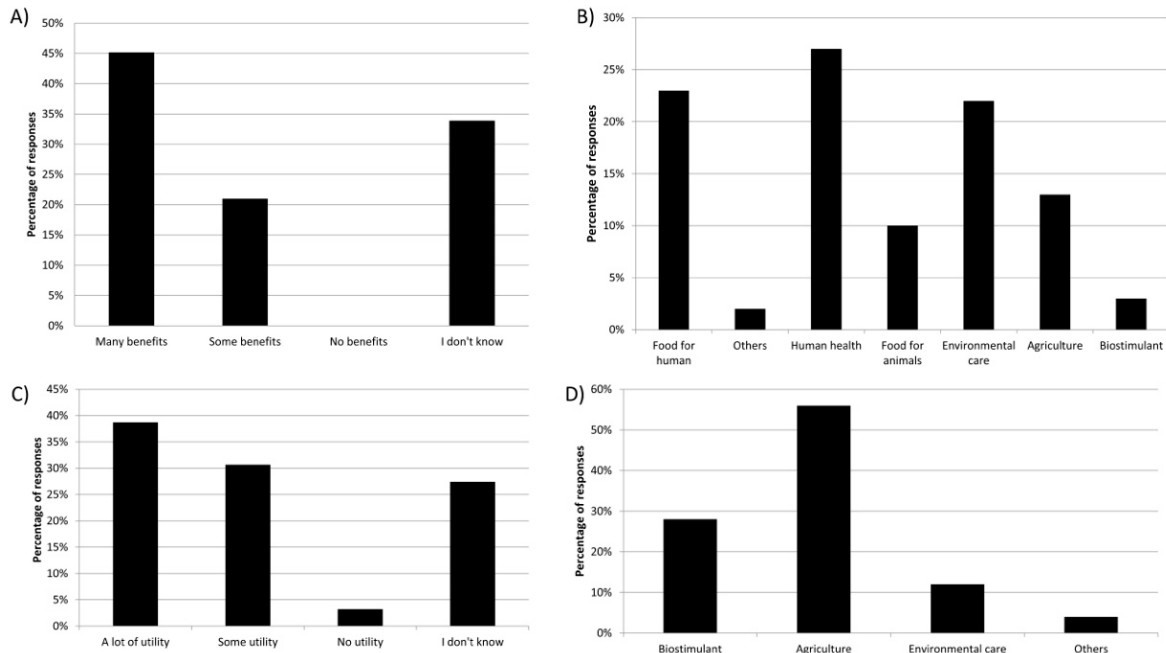

**Figure 2.** (**A**) Variety of benefits provided by microalgae for society. (**B**) Beneficial uses of microalgae for society. (**C**) Level of the utility of microalgae in agriculture. (**D**) Diversity of applications of microalgae in agriculture.

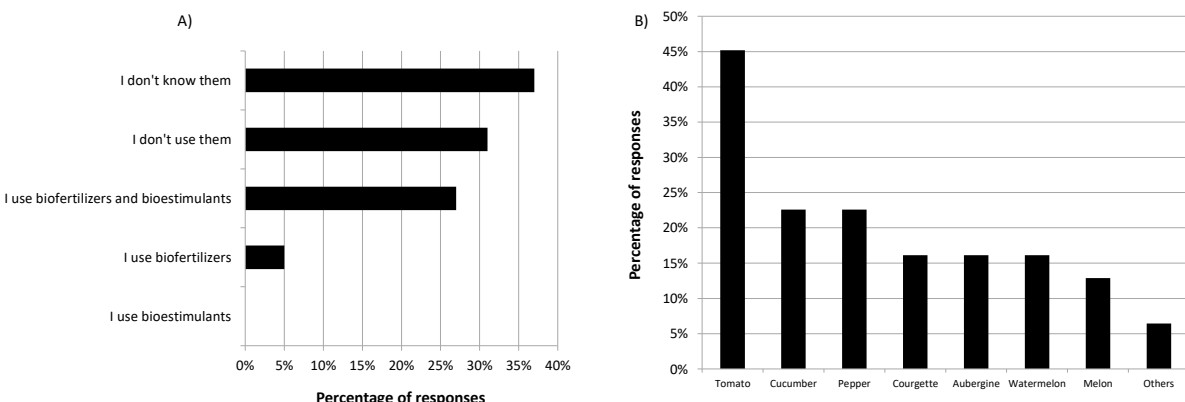

**Figure 3.** (**A**) Farmers' knowledge and use of microalgae as biostimulants and biofertilizers in agriculture. (**B**) Successful crops using microalgae.

### 3.3. Farmer's Knowledge Regarding Specific Microalgae's Applications in Agriculture

Almería's farmers identified that microalgae have very important applications in waste treatment, followed by oxygen production and stress resistance (Figure 4). Other applications identified as important in Almería's greenhouse horticulture were knowledge transfer between farmers and university, and research opportunities. A small percentage of farmers also pointed out no importance of all microalgae benefits, particularly those related to rooting and knowledge transfer between farmers and universities (Figure 4).

The Sankey diagram showed associations between specific microalgae applications and farmers' motivation to use them (Figure 5). The results indicated that a significant percentage of farmers were not able to identify any reasoning regarding the benefits of microalgae's applications to agriculture, particularly for oxygen production and waste treatment. The production of oxygen was perceived as the greatest application that microalgae can bring to greenhouse horticulture, mainly related to improvements in crop yield and environmental care. The application of waste treatment was mainly related to environmental friendliness and circular economy strategies. Stress resistance was mainly

chosen because of microalgae's capacity to mitigate the impacts of adapting to adverse climatic conditions, water limitation and crop improvement. Finally, farmers linked opportunities for research and innovation with opportunities for improvements in agricultural production, and collaboration with academia for environmental sustainability (Figure 5).

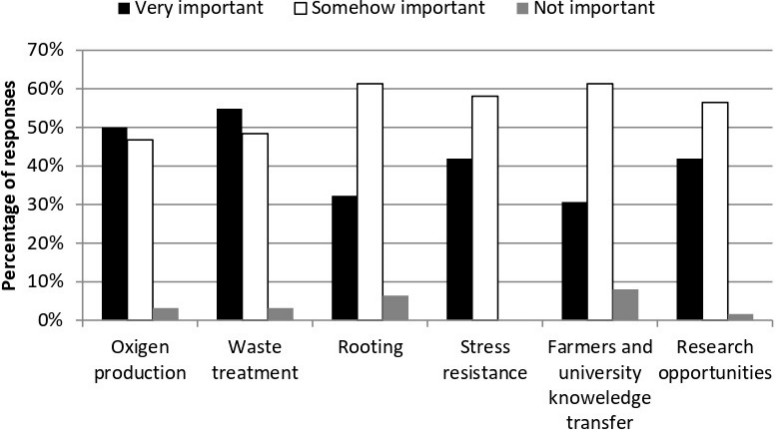

**Figure 4.** Farmers' knowledge of specific microalgae applications in Almería's agriculture.

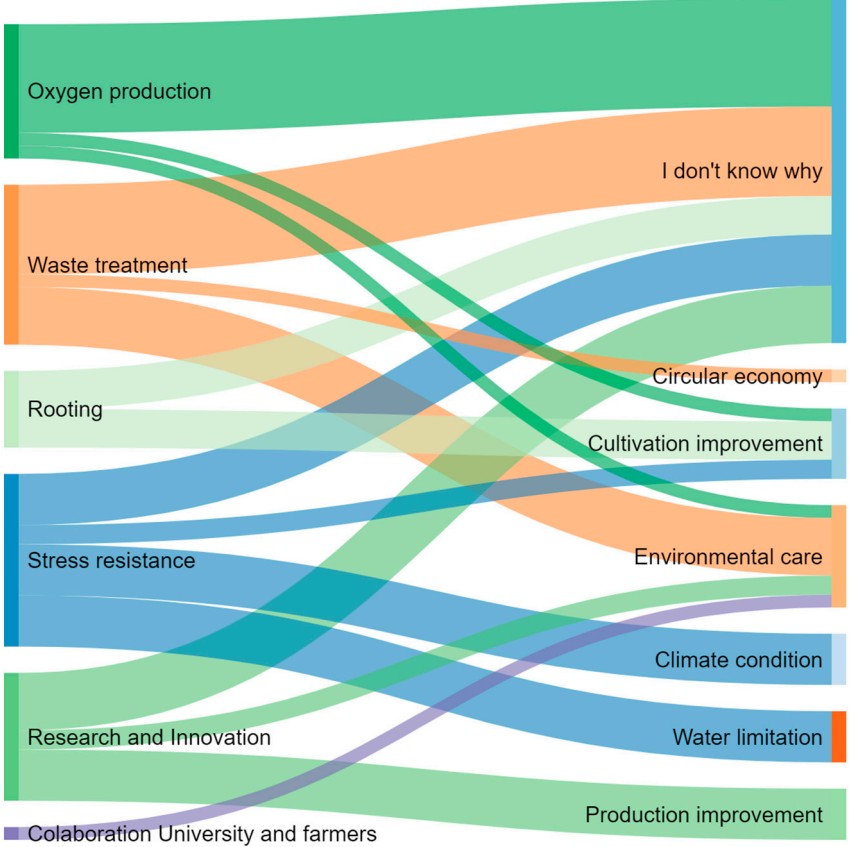

**Figure 5.** Associations between microalgae applications in agriculture and farmers' motivations.

### 3.4. Farmers' Knowledge of Microalgae's Capacity to Overcome Environmental Barriers of Almería Greenhouse Horticulture

Regarding the overcoming of barriers already existing in the greenhouse horticulture in Almería, the results show that farmers identified microalgae application as optimal to address water scarcity, followed by excessive use of chemical fertilizers and the low effectiveness of chemical pesticides (15%), the need to improve crop quality (14%) and low nutrient utilization by crops (14%). These results confirm that farmers are aware of

the most relevant barriers, thus a large interest exists in the identification of alternative technologies or strategies to solve them (Figure 6).

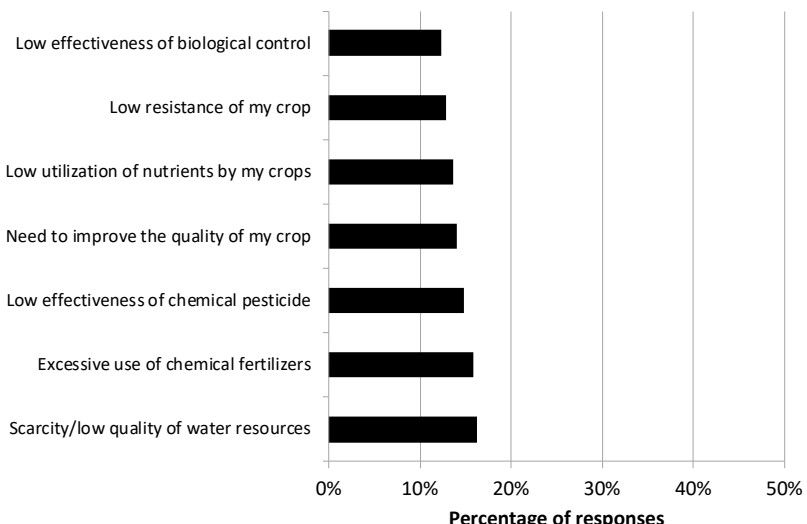

**Figure 6.** Farmers' knowledge of microalgae's capacity to mitigate environmental barriers in Almería's greenhouse horticulture.

## 4. Discussion

Current farmers' knowledge of the use of microalgae in Almería greenhouse horticulture suggests that there is basic knowledge about microalgae but a deep understanding of all their beneficial use and applications is lacking. Firstly, results indicated that farmers mainly associated the word microalgae with the marine world, and only a low percentage of farmers connected microalgae with food production and applications in agriculture. This can be explained due to the proximity to the Mediterranean Sea, which was extensively described as an explanatory factor of connectedness to nature in coastal zones [17]. Findings regarding the benefits of the use and applications of microalgae showed contradictory outcomes. On one hand, more than half of farmers identify microalgae as with potential to provide benefits to agriculture. However, we found that 34% of knowledge about microalgae's benefits, which reflects the lack of knowledge to connect microalgae-derived applications in greenhouse horticulture [18,19].

The results indicated that the most important beneficial uses of microalgae are human health, food for humans and environmental care. This finding is consistent with the overall findings of a global review of microalgae applications [19] and highlights that despite the enormous scientific advances in microalgae applications in agriculture [18,20] Almería's farmers continue to connect microalgae with human health components or as microalgae-based food, rather than as a sustainable application in agriculture.

Our findings identified the need of increasing farmers' knowledge of the specific uses of microalgae in agriculture. On one hand, approximately one-third of farmers either did not know any microalgae beneficial use or indicated that microalgae have no utility in agriculture. Additionally, although those farmers who have used the microalgae consider them as a sustainable, environmentally friendly and safe product, only 27% of farmers have used microalgae as biostimulants or biofertilizers and over 60% either do not know or have not used them. This finding suggests the need to increase consumers' knowledge of microalgae as a way to increase consumer choice and market shares of microalgae-enriched products [20].

Almería's farmers also recognized the use of microalgae as a potential solution to overcome environmental barriers in Almería's greenhouse horticulture. Specifically, farmers identified those related to addressing water scarcity, the excessive use of chemical fertilizers and the low effectiveness of chemical pesticides. This result is particularly important as it emphasizes first, environmental awareness of the agricultural sector, and secondly, the

urgency for implementing environmentally aware solutions for the long-term sustainability of Almería horticulture [1].

To advance in this direction, efficient and informative communication strategies of microalgae's benefits to reduce farmers' uncertainty are needed [20]. These strategies may involve multi-actor transdisciplinary approaches that facilitate collective work among scientists and farmers to create a culture of shared environmental responsibility among the public, private, and academic actors that constitutes the Almería greenhouse horticulture sector. A previous study on Spanish public knowledge and attitudes towards microalgae as food identified three main arguments for fostering microalgae-enriched products: (i) sustainable and environmentally friendly, (ii) nutritious and healthy, and (iii) safe. Our study found a very similar trend as the farmers' motivations to use microalgae in agriculture reflect environmental benefits such as waste treatment, oxygen production and stress resistance. Here, we highlight that all the microalgae benefits identified are strongly related to environmental friendliness such as promoting circular economy strategies, environmental care awareness, and reducing water limitation [1].

## 5. Conclusions

This research shows how Almería's farmers in SE Spain hold a limited understanding of the multiple microalgae applications for greenhouse horticulture. To fully increase and raise farmers' knowledge and acceptance around agricultural microalgae applications, the current and future needs of farmers must be taken into account to establish a relationship of trust and loyalty, as well as to plan a collective action strategy. Results confirm the need for new communication strategies that incentivizes the use and production of microalgae on a larger scale, and foster consumer choice and market shares of microalgae-enriched products.

**Author Contributions:** Conceptualization, A.J.C. and G.A.; methodology, Á.R.-N. and A.J.C.; software, Á.R.-N. and A.J.C.; validation, A.J.C. and C.G.-S.; formal analysis, Á.R.-N. and A.J.C.; investigation, Á.R.-N.; resources, Á.R.-N.; data curation, Á.R.-N.; writing original draft preparation, Á.R.-N.; writing review and editing, A.J.C. and C.G.-S.; visualization, A.J.C. and G.A.; supervision, A.J.C. and C.G.-S.; project administration, G.A. and A.J.C.; funding acquisition, G.A. and C.G.-S. All authors have read and agreed to the published version of the manuscript.

**Funding:** Open Access funding provided thanks to the SABANA project (the European Union's Horizon 2020 Research and Innovation program, grant #727874).

**Data Availability Statement:** Not applicable.

**Acknowledgments:** Funding provided thanks to the SABANA project (the European Union's Horizon 2020 Research and Innovation program, grant #727874) in addition to projects financed by Junta de Andalucía AYUDAS I+D+I EN UNIVERSIDADES Y CENTROS DE INVESTIGACION PUBLICOS. PAIDI 2020 (ALGA4FF-P20_00812; VALIMA-PY20_00800).

**Conflicts of Interest:** The authors declare no conflict of interest. The funders had no role in the design of the study; in the collection, analyses, or interpretation of data; in the writing of the manuscript; or in the decision to publish the results.

## Appendix A. Online Survey

The University of Almería is conducting research about new natural products for different agricultural crops. We would like to know your opinion, as a member of the agricultural sector, about the use of these products. Remember that this is an anonymous survey and there are no right or wrong answers, we are only interested in your opinion or experience.

*Appendix A.1. What Are Microalgae?*

1.    If I talk about microalgae, what comes to your mind? (Feel free to write whatever comes to your mind).
2.    Do you know the word microalgae? Yes/No

3. Do you know the difference between seaweed and microalgae? Yes/No
4. Do you know if microalgae have beneficial uses for agriculture?

   (a). Yes, they have many benefits
   (b). Yes, they have some benefits
   (c). Yes, they have a few benefits
   (d). No, they have no benefits
   (e). I don't know

5. If you answered yes to the previous question, list as many benefits as it comes to your mind (please separate each benefit with commas).
6. Do you know if microalgae are useful in agriculture?

   (a). Yes, they are very useful
   (b). Yes, they have some use
   (c). Yes, they have little use
   (d). No, they have no use
   (e). I don't know

7. If you answered yes to the previous question, list as many utilities as you can think of (please separate each utility with commas).

*Appendix A.2. Do You Know about Biostimulants?*

8. Do you know or have used biostimulants or biofertilizers derived from microalgae in your crops?

   (a). Yes, I know both and I use both biostimulants and biofertilizers
   (b). Yes, I know both and I use biostimulants
   (c). Yes, I know both and I use biofertilizers
   (d). Yes, I know both but I don't use either of them
   (e). No, I don't know them

9. If you have used them as biostimulants, have they given you satisfactory results?

   (a). Yes, they have given me good results in tomato, cucumber, zucchini, eggplant, bell pepper, watermelon, melon, and others.
   (b). No, they have not given me satisfactory results.
   (c). I have not observed improvements in my crop
   (d). I am not compensated for their use in relation to their cost
   (e). Microalgae'application generates additional work
   (f). For other reasons

10. If they have given you satisfactory results in other crops, could you tell us which ones?

*Appendix A.3. Possible Applications of the Use of Microalgae in Almería's Agriculture*

At the University of Almería, many studies are being carried out on the use of microalgae in agriculture. Microalgae are microscopic plant organisms that live in both fresh and salt water. Some of these applications or possible benefits of microalgae are shown in the following panel.

Take a closer look at this panel showing the benefits of microalgae.

11. Of the benefits of microalgae shown in the panel, which ones do you think are important?
12. Of the above, choose the one most important for you and indicate why.

*Appendix A.4. How Can Microalgae Help You Solve Your Problems?*

13. We show some of the difficulties or problems encountered by farmers in Almería. For each of them, please indicate how these problems affect your crops. Remember that 0 means "It is not a problem" and 10 means "It is a big problem".

    (a). Scarcity/quality of water resources.
    (b). Excessive use of chemical fertilizers.

(c). Effectiveness of chemical pesticide use.

(d). Effectiveness of using biological control.

(e). Need to improve the quality of my crop.

(f). The low resistance of my crop.

(g). Low utilization of the nutrients I provide to my crops.

(h). Indicate if there is any other problem or difficulty that has not been mentioned.

14. Do you think that microalgae can help to overcome any of these barriers? Yes/No/I don't know.

15. If yes, what environmental barrier for agriculture do you think microalgae can help with (you may select more than one option)?

(a). Scarcity/quality of water resources.

(b). Excessive use of chemical fertilizers.

(c). Effectiveness of chemical pesticide use.

(d). Effectiveness of using biological control.

(e). Need to improve the quality of my crop.

(f). Low resistance of my crop.

(g). Low utilization of the nutrients I provide to my crops.

16. Are you in favour of promoting the use of biostimulants derived from microalgae? Yes/No/I don't know.

17. If biostimulants derived from microalgae whose benefits have been demonstrated were commercialized, would you be willing to sponsor them?

*Appendix A.5. To Get to Know You Better*

18. Where do you live? Please indicate your zip code:

19. Do you come from a farming family? Yes/No.

20. Do you own land or have a lease?

(a). Yes, I am a landowner

(b). Yes, I have a land lease

(c). No.

21. How many people work in your business?

22. Could you tell me what you usually grow?

(a). Tomato

(b). Cucumber

(c). Zucchini

(d). Eggplant

(e). Pepper

(f). Watermelon

(g). Melon, Others.

23. On how much land do you grow these crops?

24. Do you have organic crops? Yes/No.

25. What is your highest level of education completed?

(a). None

(b). Primary education

(c). Secondary education

(d). High school

(e). University

(f). Others

(g). I prefer not to say.

26. Could you tell us your age?

(a). Under 20 years old

(b). Between 20 and 30 years old

(c). Between 30 and 45 years old

        (d).     Between 45 and 65 years old

        (e).     Over 65 years old.

**27.**   Gender: male/female.

**28.**   Did you find the survey comprehensible?

        (a).     Not at all comprehensible

        (b).     Not very comprehensible

        (c).     Quite comprehensible

        (d).     Very comprehensible

## Appendix B. Panel

| UTILITY | DESCRIPTION | PHOTOGRAPHY |
|---|---|---|
| Oxygen production | $CO_2$ ambient fixation and oxygen provision | |
| Waste treatment | Water purification (industrial water, slurry) | |
| Rooting | Facilitate nutrient uptake | |
| Resistance to water and temperature stress | Help withstanding water shortages and high temperature | |
| Approaching university science to agricultural activity | Offer innovative, sustainable and effective solutions for farmers | |
| Innovation and research | Favor the connection of the agricultural sector with the university | |

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
