# Peer review of "Farmers’ Knowledge and Acceptance of Microalgae in Almería Greenhouse Horticulture"

_agronomy, doi:10.3390/agronomy12112778_

Round 1

Reviewer 1 Report (Previous Reviewer 2)

The reviewers felt that the current content of the article should have some informative presentation value and help the researcher to understand the relevant situation in the region. The reviewers originally thought that some kind of theoretical framework should be used to explain this cognitive status, but the authors said that the relevant disciplinary background might not be available yet, so the reviewers had no more comments.

Reviewer 2 Report (Previous Reviewer 1)

The authors have revised their manuscript in a way that makes their research much more understandable. In my view, the paper is now ready for publication.

This manuscript is a resubmission of an earlier submission. The following is a list of the peer review reports and author responses from that submission.

Round 1

Reviewer 1 Report

The paper deals with the acceptance of microalgae by farmers, an interesting and forward-looking topic. However, from my point of view, the paper should be extensively revised. In particular, the authors should elaborate in the introduction why it is important to investigate farmers' perceptions and acceptance of this topic. The discussion should also refer more to the target group studied, i.e. farmers, and less to other stakeholder groups. In addition, it should be discussed in more detail which (further) approaches are available to increase knowledge and, if necessary, acceptance among farmers.

I also have the impression that the authors cite themselves a bit too often in relation to the total literature used.

Specific recommendations, also on the other sections, can be found in the comments to the document.

Author Response

Dear Editor(s),

Attached is a revised version of our manuscript agronomy-1859243 entitled “Farmer’s knowledge and acceptance of microalgae in Almería greenhouse Horticulture” to be considered for publication in the journal of Agronomy. We truly appreciate all suggestions from the Associate Editor and anonymous reviewers. We are confident that our revised version, in which suggested major revisions are incorporated, has been substantially improved with all detailed recommendations. Overall, below I summarize all changes and medications included in this revised version;

  • A graphic summary of the article is now included in this revised
  • Abstract and conclusion section in order to better explain the main contribution of our research.
  • All figures have been edited to increase their quality
  • Redundant text in the introduction section has been removed
  • Materials and Methods section have been substantially reworded in order to clarify all reviewers comments
  • The discussion has been substantially edited to better reflect on the innovative findings associated to our research and to not sound repetitive.

Sincerely,

Ángela Ruiz and Antonio. J Castro on behalf of all co-authors

Reviewer 2 Report

1.The Introduction section of the current manuscript contains too much redundant information. Authors need to consider refining the introduction to improve readability.

2.This manuscript lacks a review of existing research and basic theories. The authors need to fully explore the existing research to support the research logic of this study

3.The study surveyed 62 farmers online. Is the sample size sufficient to support this study? How representative of the respondents was? This needs to be further explained.

4.The description of Tables in the Result section simply repeats the data content in the table (For example, Table 1), and does not provide more information. This needs to be further improved.

5.This article is more like a descriptive statistics report. What we are seeing is a presentation of information. At present, there is a lack of exploration of the internal mechanism of this research phenomenon. The authors' previous work on information statistics and data processing has been very solid. Why not further consider exploring its internal logical mechanism? Such as adding some empirical research and heterogeneity analysis content.

6.For the innovations of this paper, the authors lack a summary in the manuscript. What are the outstanding innovations and marginal contributions of the current research compared with existing research?

7.In this manuscript, the pictures are low-pixel and somewhat blurry. The authors need to further enhance the image pixels to make it more clear. In particular, the drawing of Figure 2 is too simple and shoddy.

Author Response

(The authors gave the same response as above.)

Round 2

Reviewer 2 Report

In my personal opinion, if the article only presents the results of 62 respondents and only a simple descriptive presentation of the results, then the innovation of this study and its appeal to other readers worldwide is not sufficient to be published in this journal.

If the research design of the article had considered revealing the mechanisms of what influences are affecting farmers' perceptions of microalgae and how, then it would have been possible to suggest universal patterns about farmers' perceptions of microalgae. Throughout the article, however, it was not so designed, and so no more actionable policy recommendations could be made.

The introduction to the article should have included a more in-depth analysis of what research has been done on farmers' perceptions of microalgae, what research understanding has been achieved, and what areas are still weak, rather than the current one that remains on the importance of microalgae, which is certainly important, but is far from the topic of this article.